# Enhanced Photoelectrochemical Performance Using Cobalt-Catalyst-Loaded PVD/RF-Engineered WO_3_ Photoelectrodes

**DOI:** 10.3390/nano14030259

**Published:** 2024-01-25

**Authors:** Mansour Alhabradi, Xiuru Yang, Manal Alruwaili, Hong Chang, Asif Ali Tahir

**Affiliations:** 1Environment and Sustainability Institute, University of Exeter, Penryn TR10 9FE, UK; xy328@exeter.ac.uk (X.Y.); ma942@exeter.ac.uk (M.A.); a.tahir@exeter.ac.uk (A.A.T.); 2Department of Physics, Faculty of Science, Majmaah University, Majmaah 11952, Saudi Arabia; 3Department of Physics, Faculty of Science, Jouf University, Sakaka 42421, Saudi Arabia; 4Faculty of Environment, Science and Economy, University of Exeter, Exeter EX4 4QF, UK; h.chang@exeter.ac.uk

**Keywords:** photoelectrochemical (PEC), cobalt nanoparticles, PVD/RF method, tungsten oxide (WO_3_) thin film

## Abstract

Critical to boosting photoelectrochemical (PEC) performance is improving visible light absorption, accelerating carrier separation, and reducing electron–hole pair recombination. In this investigation, the PVD/RF method was employed to fabricate WO_3_ thin films that were subsequently treated using the surface treatment process, and the film surface was modified by introducing varying concentrations of cobalt nanoparticles, a non-noble metal, as an effective Co catalyst. The results show that the impact of loaded cobalt nanoparticles on the film surface can explain the extended absorption spectrum of visible light, efficiently capturing photogenerated electrons. This leads to an increased concentration of charge carriers, promoting a faster rate of carrier separation and enhancing interface charge transfer efficiency. Compared with a pristine WO_3_ thin film photoanode, the photocurrent of the as-prepared Co/WO_3_ films shows a higher PEC activity, with more than a one-fold increase in photocurrent density from 1.020 mA/cm^2^ to 1.485 mA/cm^2^ under simulated solar radiation. The phase, crystallinity, and surface of the prepared films were analysed using X-ray diffraction (XRD), X-ray photoelectron spectroscopy (XPS), and Raman spectroscopy. The PVD/RF method, scanning electron microscopy (FE-SEM), and high-resolution transmission electron microscopy (HR-TEM) were employed to assess the surface morphology of the fabricated film electrode. Optical properties were studied using UV–vis absorbance spectroscopy. Simultaneously, the photoelectrochemical properties of both films were evaluated using linear sweep voltammetry and electrochemical impedance spectroscopy (EIS). These results offer a valuable reference for designing high-performance photoanodes on a large scale for photoelectrochemical (PEC) applications.

## 1. Introduction

The next generation of hydrogen, as a sustainable energy source with a clean and carbon-free production process through photoelectrochemical water splitting [1], is a promising solution to the future global energy demand and need for environmental protection. In photoelectrochemical (PEC) water splitting, a semiconducting material is used to separate water into H_2_ and O_2_ using abundant and renewable resources, such as sunlight and water [2,3]. Fujishima and Honda [4] reported the first effective demonstration of solar water splitting using a TiO_2_ semiconductor, which sparked significant research interest and contributions that have persisted to the present day. Consequently, various semiconductors have been investigated, researched, and reported as active materials used for solar water-splitting reactions [5,6,7]. Notably, metal oxides’ well-matched optical, electrical, and stability properties have been shown to be useful electrode materials in PEC water-splitting systems.

Due to its favourable physical and chemical properties [8], tungsten oxide (WO_3_) has emerged as a viable candidate among the numerous metal oxide materials used in PEC water splitting. In 1976, Hodes et al. [9] were the first to demonstrate the effective use of WO_3_ in solar water splitting. Significant research contributions have been reported since then. It was discovered that WO_3_ can absorb a maximum of 12% of the solar spectrum with a band gap between 2.5 and 2.8 eV [10]. Theoretically, based on its light absorption capacity, as much as 4.8% of the used solar energy could be converted into hydrogen [11]. In addition, chemical stability, extended durability, non-toxicity, cost-effectiveness, and transparency to visible light are some of WO_3_’s advantages [12].

However, WO_3_ has numerous limitations that influence its charge generation, separation, and transportation properties, decreasing its ability to carry out water splitting [13]. Therefore, the structural, optical, and electrochemical properties of WO_3_ must be modified to increase its water-splitting efficacy. Moreover, due to its bandgap, which permits the absorption of only a small percentage of incident sunlight, WO_3_ cannot be considered an optimal material for water splitting. Therefore, bandgap modification (i.e., narrowing the bandgap) is required to enable WO_3_ to absorb a sufficient amount of solar light to enhance the efficacy of PEC water splitting. As a result, the charge carriers’ rate of recombination is rapid, while the transit and trapping of charge carriers proceed slowly on the surface [14].

Several approaches have been used to improve the photoelectrochemical (PEC) properties of tungsten trioxide (WO_3_). These include modifying the morphology of WO_3_, introducing transition-metal dopants, loading noble metals onto the WO_3_ surface, sensitising the surface, and constructing composite materials. Doping and co-doping have been identified as useful techniques for modifying the electronic band structure and improving the photoelectrochemical (PEC) properties of tungsten trioxide (WO_3_) [15]. The literature has reported the doping of many metals, including titanium (Ti), zinc (Zn), dysprosium (Dy), tellurium (Te), tantalum (Ta), vanadium (V), copper (Cu), silver (Ag), cerium (Ce), magnesium (Mg), molybdenum (Mo), and nickel (Ni), into tungsten trioxide (WO_3_) [16,17,18,19,20,21,22,23,24,25,26,27]. The primary aims involved in employing the Co catalyst approach for impurity implantation on the surface of a photocatalyst in the context of photoelectrochemical (PEC) water splitting are to broaden the spectral response of these materials to encompass visible light through the augmentation of the material’s light-harvesting capabilities, facilitate the charge transfer processes, and provide active sites within these materials [28]. Furthermore, the incorporation of Co catalysts has been shown to decrease the overpotential associated with the hydrogen evolution (HER) and oxygen evolution (OER) processes, hence enhancing the long-term stability of these reactions. Pawar et al. conducted an evaluation of the surface plasmon resonance (SPR) impact resulting from the loading of silver (Ag) and nickel (Ni) nanoparticles (NPs) onto thin film surfaces of lanthanum iron oxide (LaFeO_3_). The Ag and Ni/LaFeO_3_ samples demonstrated enhanced light absorption, leading to an improvement in current density in comparison to the bare LaFeO_3_ sample [29,30].

The water-splitting properties of WO_3_ photoanodes have been observed to be significantly affected by the morphology, microstructure, phase composition, and contact area between the film and charge collector, as has been demonstrated through various techniques such as the sol–gel method [31], chemical vapour deposition (CVD) [32], physical vapour deposition (PVD) [33], and anodisation [34].

In the present work, we employed a PVD/RF method to fabricate WO_3_ thin films that were subsequently subjected to a surface treatment process and then loaded with cobalt nanoparticles on their surface to enhance their PEC performance. Various concentrations ranging from 1 mM to 4 mM solutions were employed to obtain Co/WO_3_ films. The photoanodes composed of 3 mM WO_3_/Co films demonstrated the highest photocurrent at 1.485 mA/cm^2^ at 1.23 V vs. RHE. This enhanced performance can be attributed to the elevated donor density, measured at 4.6 × 10^21^ cm^−3^, observed within the films. To the best of our knowledge from the existing literature, there is no prior research reporting the impact of cobalt nanoparticles loaded on a WO_3_ surface for the improvement of photoelectrochemical (PEC) performance.

## 2. Experimental Section

### 2.1. Fabrication of RF-Sputtered Nanostructured WO_3_ Thin Films

The fabrication process for WO_3_ thin films used as photoanodes for photoelectrochemical (PEC) applications involved the PVD/RF technique. This technique employed pure tungsten (W) metal targets (99.95% purity, Moorfield Nanotechnology) with dimensions of 3″ × 1/16″. This target was deposited onto glass substrates coated with fluorine-doped tin oxide (FTO, NSG TEC 5, Pilkington, Saint Helens, UK) measuring 1 by 2 cm. The deposition was carried out using RF magnetron sputtering at room temperature, as depicted in Figure 1. Before depositing the thin layer of film, the FTO substrates underwent a cleaning process involving ultrasonication with deionised water and acetone. Subsequently, they were stored in ethanol prior to use. During the deposition process, the distance between the FTO substrate and the target (ds-t) was maintained at 15 cm, and the deposition angle (α) was set at 30° to achieve the desired nanostructured morphology. The sputtering was conducted in a vacuum environment with a working pressure of 0.3 Pa, with 20 sccm (standard cubic centimetre per minute) of 99.99% pure argon (Ar) gas introduced into the deposition chamber. The deposition rate and film thickness were monitored using a quartz crystal microbalance (QCM). Before depositing tungsten on the substrate, a pre-sputtering step was performed for 5–10 min in argon (Ar) plasma, with a shutter protecting the surface from contamination. The sputtering power was maintained at 75 W for the tungsten target. Throughout the deposition, the substrate was rotated at a speed of 10 rpm to ensure uniform thickness. To optimise the PEC performance, various samples were prepared and tested. The as-prepared samples underwent high-temperature annealing in a muffle furnace at 600 °C for 2 h, with a heating rate of 5° per minute, to create a well-structured nanostructured photoanode. Finally, the samples were allowed to cool naturally to room temperature before undergoing characterisation or experimentation.

### 2.2. Synthesis of the WO_3_ Photoanode with Cobalt Nanoparticles

The WO_3_ films were submerged in a solution containing 0.7 mM of sodium citrate, along with different concentrations of cobalt nitrate (1, 2, 3, and 4 mM) [35]. This immersion process was carried out at a temperature of 100 °C for a duration of 2 h. Following this, the WO_3_/Co films underwent a thorough washing with deionised water and were left to dry in the open air, as shown in Figure 1. These prepared films were subsequently employed to evaluate their performance in photoelectrochemical (PEC) testing. 

### 2.3. Characterisations

The crystal structure and phases of the fabricated Co/WO_3_ films were examined using a Bruker D8 X-ray diffractometer, which used monochromatic Cu-K radiation with a wavelength of 0.154 nm. Raman spectra were acquired using a confocal Raman microscope (Alpha300, WITec GmbH, Germany). For Raman spectroscopy, a visible laser with a wavelength of 532 nm and an output power of 15 mW was employed to excite the sample, and the backscattered light was captured using a 50-objective lens. The analysis involved averaging 10 individual spectra. 

To study the surface morphology of the photoelectrodes and analyse the elements present in the photoelectrode films, scanning electron microscopy (FE-SEM) with a TESCAN VEGA3 instrument (Oxford instrument elemental analysis, UK) and Energy Dispersive Spectroscopy (EDS) from Oxford Instruments were used. Additionally, the structural characterisation involved high-resolution transmission electron microscopy (HR-TEM), selected area electron diffraction (SAED), and scanning transmission electron microscopy (STEM). The film thickness was determined using an FEI Nova 600 Nanolab FIB/SEM system (Thermo Fisher Scientific, Loughborough, UK).

To assess the optical properties of the coated photoelectrodes, a PerkinElmer LAMBDA 1050 UV/vis/NIR spectrophotometer (PerkinElmer, Waltham, MA, USA) was employed, covering a wavelength range of 200–850 nm. For X-ray photoelectron spectroscopy (XPS) measurements, a Thermo Fisher Scientific NEXSA (Thermo Fisher Scientific, Loughborough, UK) spectrometer was used. This involved the use of a micro-focused monochromatic Al X-ray source (72 W) to analyse the materials across an area of approximately 400 microns. 

### 2.4. Electrochemical Measurements

The photoelectrochemical (PEC) performances were conducted using a Newport 66902, 300 W xenon lamp (Newport Spectra-Physics Ltd., Cheshire, UK) equipped with an AM 1.5 filter within a three-electrode system. In this setup, platinum served as the counter electrode, Ag/AgCl in 3M KCL was the reference electrode, and the working electrode was the FTO-coated thin film. These experiments were carried out on a Metrohm Autolab (PGSTAT302N) workstation, simulating 1 sun condition (100 mW/cm^2^). The electrolyte used was a 1.0 M NaOH aqueous solution with a pH of 13.6. Current–voltage measurements for all working electrodes were taken within a potential range of (−0.3 V to +0.7 V) at a scan rate of 0.01 V/s. Throughout the PEC measurements, each electrode received illumination from the front, and a consistent working electrode area of 1.0 cm^2^ was maintained. The working electrode potential was calculated relative to a reversible hydrogen electrode (RHE) using the Nernst formula (Equation (1)):(1)$$E_{RHE}=E_{Ag/AgCl}+E_{Ag/AgCl}^{0}+0.0591\;V\times pH\;\;\;(E_{Ag/AgCl}^{0}=0.1976\;V\;vs.\;NHE\;at\;25\;{}^{\circ}C)$$

Additionally, the Mott–Schottky relationship was used to determine the flat band potential (*E_fb_*) of the pure WO_3_ and the Co-loaded WO_3_ thin films, employing the following equation (Equation (2)):(2)$$\frac{1}{C^{2}}=\frac{2}{e\varepsilon \varepsilon_{0}A^{2}N_{D}}\left(E-E_{fb}-\frac{KT}{e}\right)$$

In this equation, *C* represents the space charge capacitance, *e* denotes the electron charge (1.6022 × 10^−19^ C), *ε* is the dielectric constant of WO_3_ (20) [36,37], *ε*_0_ represents the dielectric constant of the semiconductor and the permittivity of free space (8.854 × 10^−12^ F/m), A stands for the area of the thin film, *N_D_* is the dopant density, *E* and *E_fb_* are the applied and flat band potential, respectively, and *K_b_* and *T* denote the Boltzmann constant (1.38 × 10^−23^ J/K) and absolute temperature, respectively.

## 3. Result and Discussion

### 3.1. XRD Structural and Raman Characterisation

The X-ray diffraction patterns of the pristine and loaded tungsten oxide samples, as shown in Figure 2, were examined to study their respective phases. The diffraction peaks observed in the analysis perfectly correspond to the crystal structure of monoclinic WO_3_ (as described by JCPDS card No. 032-1395) with specific lattice constants of a = 7.309 Å, b = 7.522 Å, and c = 7.678 Å [38]. The main diffraction peaks observed at 23°, 23.5°, and 24.3° correspond to the (002), (020), and (200) crystallographic planes of the WO_3_ monoclinic phase, respectively. The absence of any peaks related to impurities or secondary phases provides strong evidence that the samples are in a pure, single phase. The well-defined peaks (marked in red) are associated with the tetragonal SnO_2_ crystal structure (referencing JCPDS No. 077-0451), which originates from the FTO substrate. The interconnected crystalline arrangement of WO_3_ particles plays a crucial role in creating a substantial interface between the electrolyte and the film, as well as facilitating efficient electron transport within the film. These factors are expected to enhance the effectiveness of the photoelectrochemical (PEC) water oxidation process. This is because when WO_3_ is photoexcited, the electron–hole pairs generated have a reduced likelihood of recombining, allowing them to participate in the water oxidation reaction on the WO_3_ surface more effectively [39]. After Co doping, no diffraction peaks were detected in the Co/WO_3_ (3 mM) film, primarily attributable to the minimal quantity of Co loaded on the surface.

Raman spectroscopy proves to be a valuable tool for understanding the crystal structure and bonding characteristics of the materials. In Figure 3, the Raman spectrum reveals five distinct vibrational bands within the prepared samples. These vibrational modes collectively affirm the presence of monoclinic-phase WO_3_. Furthermore, the absence of secondary-phase and impurity-related peaks underscores the purity of the prepared samples.

A minor peak, observed at a lower wavenumber (132 cm^−1^), corresponds to the lattice vibrational modes of WO_3_. Peaks detected at 270 cm^−1^ and 326 cm^−1^ are associated with the δ(O–W–O) bending vibrations in the WO_3_ samples. Simultaneously, the υ(O–W–O) stretching vibrations are characterised by two prominent Raman peaks at higher wavenumbers, ranging from 700 to 850 cm^−1^. In the case of Co-loaded samples, a slight shift in peak position is evident, possibly due to the creation of oxygen vacancies within the WO_3_ crystal matrix [40]. In contrast to pure WO_3_, the intensities of the Raman peaks in the Co/WO_3_ thin films reduced as the concentration of the doping increased compared with the undoped. This reduction in intensity is attributed to the substitutional effect, where the doping atoms replace atoms in the host matrix. This observation aligns with findings previously reported in the scientific literature [41].

### 3.2. UV-DRS Spectroscopic Analysis 

UV–vis diffuse reflectance spectroscopic (UV-DRS) analysis was used to determine the optical absorption characteristics of unmodified and Co/WO_3_ films. Figure 4a illustrates the UV–vis absorption spectra for different Co concentrations loaded onto the WO_3_. Notably, there is an increased absorption in the wavelength range (350–500 nm). In addition, enhanced light absorbance is attributed to the presence of Co on the bare film, although it should be relatively minor given that more incident light is absorbed with higher Co loading. It is worth mentioning that the absorption edges remain almost the same after Co particle loading, signifying that Co is deposited on the surface rather than being incorporated into the WO_3_ lattice.

The band gap of the optimised loaded Co nanoparticles was determined by analysing the Tauc plots, as seen in Figure 4b. The respective band gaps were found to be 2.5 and 2.76 eV for the unloaded and Co/ WO_3_ films, respectively. On the other hand, more Co loading (4 mM) reduced light absorption due to scattering from agglomerated Co on the surface, obstructing more active sites of the photocatalyst. These observations highlight the significant impact of Co catalyst loading on photocatalytic absorption. Higher Co catalyst concentrations can shield incident light, leading to reduced photocatalytic activity [42,43,44]. Therefore, based on the influence of the absorption spectra, a concentration of 3 mM was identified as the optimal level for Co loading since it exhibits maximum absorption, justifying further analysis of the PEC performance.

### 3.3. Surface Morphology and Elemental Analysis

The surface structure and elemental composition of bare WO_3_ and Co/WO_3_ films with different Co concentrations were analysed using field emission scanning electron microscopy (FE-SEM) and FE-SEM combined with an Energy Dispersive X-ray Spectroscopy (EDX) system, as shown in Figure 5a–g. The top-view scanning electron microscopy (SEM) images of several films are shown in Figure 5a–f. These films include untreated WO_3_ as well as WO_3_ films treated with varying concentrations of Co solution. Figure 5a displays the unaltered WO_3_ film, exhibiting a porous structure characterised by distinct grains and a uniformly smooth surface. The original spatial structure of WO_3_ remains intact, but there is a more noticeable and rougher adhered-like structure on the tail end of the WO_3_ particles, as shown in Figure 5a. When a small amount of Co (1 mM) is added to the bare WO_3_ surface, as is shown in Figure 5c, there are changes in the shape of the grains, but the changes are not efficient enough to reach the optimal. Moreover, when increasing the amount of Co to 2 mM, it is observed that a thin layer on the surface of the particle is formed, as shown in Figure 5c.

When Co (3 mM) is loaded onto the WO_3_ surface, yielding the highest photocurrent, the surface remains relatively smooth; the Co particles are evenly distributed, and the particle connections are enhanced, as shown in Figure 5d. Figure 5e demonstrates that an excess in the Co catalyst causes uneven primary particle sizes, hides edges beneath a thick layer, and gives rise to larger and irregular grain sizes due to particle aggregation. Figure 5f shows a cross-sectional SEM image of Co (3 mM) loaded onto the surface of the WO_3_ film.

In contrast to the structure of the unmodified film, the Co/WO_3_ thin film exhibits a structure conducive to achieving a moderate pore size and promoting active contact sites with the electrolyte. This configuration facilitates the separation and movement of charges, leading to an enhancement in photocurrent density. Therefore, the presence of Co particles on the WO_3_ surface enhances the inherent photocatalytic activity of the material. The EDS analysis of Co/WO_3_, as seen in Figure 5g, provides confirmation of the existence of the W, Sn, Si, and Co components. The elemental surface scanning images reveal a homogeneous dispersion of cobalt (Co) particles on the surface of the WO_3_ thin films.

To further examine the developed Co/WO_3_ film (3 mM) crystals, transmission electron microscopy (TEM) measurements were carried out, as shown in Figure 6. Figure 6a displays a low-magnification TEM image of the Co/WO_3_ film (3 mM) sample. It illustrates similar plate-like structures with an average nanoparticle diameter of approximately 50 nm. As presented in Figure 6b, most of the Co particles that have been decorated on the WO_3_ surface exhibit a shape that can be described as squircular, which means they have well-rounded corners and a nearly square or circular appearance. Figure 6c reveals a lattice structure with parallel fringes, and the spacing between these fringes measures 0.376 nm, corresponding to the (020) plane of monoclinic WO_3_. In Figure 6d, STEM images of the WO_3_ loaded with Co are presented, revealing a consistent and even dispersion of Co particles over the surface of WO_3_. Furthermore, the SAED pattern seen in Figure 6e exhibits diffraction planes, such as (020), (200), and (222), which provide evidence for the presence of the monoclinic crystal structure in WO_3_.

### 3.4. X-ray Photoelectron Spectroscopic (XPS) Analysis

The X-ray photoelectron spectroscopy (XPS) technique was employed to investigate the chemical composition and oxidation state of metal ions in both the pure and Co/WO_3_ thin films that were synthesised, as shown in Figure 6. The calibration of binding energies was conducted using carbon (C 1s = 283.4 eV) as the reference for all measurements. In Figure 7a, the survey scans of unloaded and Co-loaded WO_3_ thin films fabricated through the PVD process are presented. The scans cover the entire energy range of 0–1000 eV and indicate the presence of W, O, and C elements on the surface of these two samples, where the C may be due to the adsorbed carbon species in the air. Furthermore, it has been previously established that the Sn signal originates from the substrate of the FTO [45]. In addition, there are evident signs of cobalt (Co) element present in the tungsten oxide (WO_3_) thin films. Figure 7b,c displays the deconvoluted core-level spectra of W 4f for both WO_3_ and Co/WO_3_ thin films. Figure 6b,c presents the core-level XPS spectrum of W 4f, revealing the existence of two distinctive peaks attributed to 4f_7/2_ and 4f_5/2_ located at 35.5 eV and 37.6 eV, respectively, for both samples [46]. The presence of these peaks at their respective energy levels confirms the presence of W elements in the sample in the W6+ oxidation state. In comparison, WO_3_ loaded with Co resulted in a reduction in the intensity of both energy levels, as shown in Figure 7c. Figure 7d,e displays the deconvoluted peaks seen in the O 1s spectra of both pure and Co-loaded WO_3_ thin films. The XPS analysis of the O 1s spectrum in the pristine film revealed the presence of two distinct binding energies. The first binding energy, measured at 530.2 eV, corresponds to lattice oxygen (O_L_). The second binding energy, observed at 531.3 eV, indicates the existence of oxygen vacancies (O_v_). In contrast, the Co/WO_3_ thin films also show similar binding energies at 530.3 eV and 532.2 eV, corresponding to lattice oxygen (O_L_) and oxygen vacancies (O_v_), respectively. Importantly, the outcome of the study confirmed the existence of oxygen vacancies (O_v_) in the Co/WO_3_ thin films, which exhibited an increase, as shown in Figure 7e. This rise suggests that the presence of Co may have contributed to the generation of additional oxygen vacancies on the surface of the film. Oxygen vacancies are known to have a substantial impact on enhancing the photocatalytic capabilities via their influence on the interface transfer performance [47]. The cobalt 2p core-level spectra exhibit two distinct peaks, which correspond to the 2p_3/2_ and 2p_1/2_ orbitals. These peaks are seen at binding energies of 781.43 and 797.42, respectively, as shown in Figure 6e. The observed discrepancy in binding energy between these two levels was determined to be 15.9 eV, indicating the division of the spin–orbit doublet of Co^2+^. The cobalt 2p spectra may be deconvoluted, revealing the presence of two minor satellite peaks at 785.7 and 802.6 eV, which correspond to the 2p_3/2_ and 2p_1/2_ energy levels, respectively. The confirmation of the coexistence of tetrahedral Co^2+^ and octahedral Co^3+^ was achieved by analysing the deconvoluted 2p spectra, which align with the cobalt 2p spectra [48].

### 3.5. Photoelectrochemical Measurements

To analyse the photoelectrochemical (PEC) efficiency of untreated and Co/WO_3_ thin films, the study conducted LSV measurements on the prepared samples in both dark and light conditions, as shown in Figure 8. The results presented in Figure 8a indicate that the dark currents were notably low when compared with the corresponding photocurrent density values observed in the presence of illumination. Upon irradiation, the bare WO_3_ exhibited a photocurrent of 1.020 mA/cm^2^ at 1.23 V vs. RHE, which is higher than that reported recently [49]. The enhancement in photocurrent density in exposed WO_3_ films is evident from the results depicted in Figure 8a upon the addition of Co NPs. The photocurrent density was shown to gradually improve upon the loading of Co particles with concentrations of 1, 2, and 3 mM, reaching values of 1.135, 1.337, and 1.485 mA/cm^2^, respectively, at a potential of 1.23 V versus RHE. The observed phenomenon may be attributed to the enhancement in light absorption and the increased number of active sites on the surface of the photocatalyst, as seen in the absorption spectra. Nevertheless, when a Co solution with a concentration greater than 4 mM was loaded, a significant reduction in the photocurrent density value of 0.846 mA/cm^2^ was found. The observed decrease in photocurrent density can be related to the increased loading impact of cobalt particles on the film’s surface, resulting in an inner filter effect [29]. This phenomenon hinders the accessibility of active sites and diminishes the interaction between the photoanode and electrolyte, leading to rapid bulk recombination. Table 1 presents the performance of our synthesised tungsten oxide (WO_3_) films loaded with Co NPs compared to those reported in the literature. The present study demonstrates that the photocurrent value obtained for the optimal load is higher than the one reported in the literature compared with other Co catalysts. Based on the aforementioned findings, it can be inferred that the increase in photocurrent density can be attributed to the ability of Co NPs to facilitate the efficient separation and transfer of carriers.

Employing electrochemical impedance spectroscopic (EIS) analysis has been widely recognised as a valuable technique for studying the kinetics of photoelectrochemical processes. To obtain a deeper understanding of electron transport dynamics and elucidate the disparities in interfacial properties between the untreated and Co/WO_3_ (3 mM) thin films, the electrochemical impedance spectroscopy (EIS) technique was employed to analyse the photoelectrodes, as seen in Figure 8b. The Nyquist plots in Figure 7b illustrate the charge transfer occurring at the interface between the photoanode and electrolyte. This charge transfer may be accurately represented by an equivalent circuit that includes a series resistance (R_s_), a charge transfer resistance (R_ct_), and a constant phase element (CPE), as shown in the inset of Figure 8b. The findings of the electrochemical impedance spectroscopy (EIS) investigation indicate that the Co/WO_3_ (3 mM) system exhibits lower R_s_ and R_ct_ values (19.7 and 47.8 ohm, respectively) than the WO_3_ system (27.6 and 86.6 ohm, respectively). This observation suggests that the presence of the Co NPs enhances the efficiency of charge transfer at the interface between the photoanode and electrolyte. The bode phase plots depicted in Figure 8c illustrate the frequency peaks associated with the charge transfer process occurring at the interface of each photoanode in the EIS spectra. The impedance semicircle of Co/WO_3_ (3 mM) thin films has a lower maximum oscillation frequency (f_max_) than the maximum oscillation frequency of WO_3_. When examining the relationship between electron lifetime (τ_e_) and f_max_, it is seen that the lifetime of photoelectrons in the Co/WO_3_ (3 mM) thin films is greater than that of WO_3_, as stated in the following equation [52]: (3)$$\tau_{e}=\frac{1}{2\pi f_{\left(\max\right)}}$$

The bode plot was used to determine the highest frequency peaks (f_max_) of both bare Co/WO_3_ (3 mM) thin films. The findings indicate that there is a decrease in the frequency peak (f_max_) for the WO_3_ and Co/WO_3_ (3 mM) thin films, shifting from 850 Hz to 457 Hz, respectively. This finding provides the first hint of an extended electron lifetime in the formation of loaded Co NPs on the surface of the WO_3_ thin films. The respective values of τ_e_ for WO_3_ and Co/WO_3_ (3 mM) thin films are 1.8 and 3.5 ms, respectively.

Figure 8d illustrates the Mott–Schottky plots of the WO_3_ and Co/WO_3_ (3 mM) samples in their as-prepared state. The confirmation of tungsten oxide’s n-type character was supported by the positive slopes seen in the plots [53]. The calculated E_fb_ and N_D_ values are −0.48 V vs. Ag/AgCl and 3.6 × 10^21^ cm^−3^ for WO_3_ compared with −0.57 V vs. Ag/AgCl and 4.6 × 10^21^ cm^−3^ for Co/WO_3_ (3 mM) thin films, respectively. The values of E_fb_ were determined by identifying the x-axis intercepts of the extrapolated linear portions of the plots. On the other hand, the values of N_D_ were calculated based on the slopes of the plots, which correspond to the Mott–Schottky relationship, as described in Equation (3). The enhancement in donor concentration may be achieved with the use of Co NPs, hence facilitating charge transfer in WO_3_. This could be attributed to the formation of a Schottky barrier at the interface between the WO_3_ and Co nanoparticles, which serves as an electron reservoir within WO_3_. When a metal comes into close contact with a semiconductor, a Schottky barrier is typically formed, leading to band bending at the semiconductor’s interface. This interaction requires the electrons to possess higher energy to overcome the barrier created by this junction, as has been reported in the literature [54,55]. The efficient separation and transportation of charge, along with the improved absorption of visible light, are believed to be key factors in the improved photoelectrochemical (PEC) performance observed in the Co/WO_3_ thin films [56]. Nevertheless, an increased loading quantity of Co particles enables their function as recombination centres, subsequently leading to the aggregation and creation of a more substantial barrier layer. Consequently, this impedes the transfer of charges, ultimately resulting in a reduced photocurrent density, as seen in the SEM pictures.

Combining the experimental findings with the aforementioned characterisation analyses allows for the mechanism underlying the photocatalytic activity of the Co/WO_3_ films to be explained. Although the band gap energies of the Co/WO_3_ thin films and WO_3_ are similar (2.5–2.7 eV), it can be observed that the Co/WO_3_ thin films display somewhat greater negative conduction band (CB) potentials than WO_3_. The process involves the deposition of Co particles onto the photocatalyst’s surface, facilitating the transport of excited electrons from the conduction band of WO_3_ to the Co particles situated on the surface. The aforementioned transfer serves to inhibit recombination inside the bulk film. In contrast, the photo-induced holes within the valence band persist on the photocatalyst. As a result, the Co particles gather electrons that actively engage in the reduction process, whilst the holes disperse towards the surface of the photocatalyst and play a role in oxidation reactions, thus facilitating the entire process of the water-splitting reactions.

## 4. Conclusions

Using a high-performing and easily tuneable PVD/RF method, both pristine WO_3_ and Co/WO_3_ thin films were successfully fabricated; this was followed by a thermal oxidation step that yielded films with a highly crystalline structure. The incorporation of cobalt nanoparticles (Co NPs) into tungsten oxides (WO_3_) thin films led to improved performance and absorption spectra, with the extent of enhancement being dependent on the amount of cobalt employed. The inclusion of Co^2+^ ions in the solution resulted in additional enhancements to the WO_3_ thin film, including the reduction in recombination rates and the facilitation of electron transfers, as shown in EIS results. The performance of the photocatalysts was significantly influenced by the concentration of Co loading, with the most optimal performance seen when the Co loading was 3 mM. The observed negative shift holds practical importance for photoelectrochemical water splitting, as the incorporation of Co/WO_3_ resulted in a slight reduction in the band gap of the unmodified film, as evidenced by an observable downward shift in the potential scan rate on the Mott–Schottky plot. The enhanced photoelectrochemical (PEC) performance of the WO_3_ thin film was ascribed to the increased presence of oxygen vacancies and the enhanced alignment of primary WO_3_ particles, which resulted from the addition of Co at a concentration of 3 mM. Nevertheless, as the loading quantity exceeded 4 mM, there was a noticeable decline in photocurrent density. This decline might be attributed to the agglomeration of cobalt and the rapid recombination of charge carriers. In addition, the use of Co catalyst nanoparticles in a cost-effective and uncomplicated manner presents compelling evidence in favour of employing photoanodes. This finding establishes their potential suitability for other photocatalysts.

## Figures and Tables

**Figure 1 nanomaterials-14-00259-f001:**
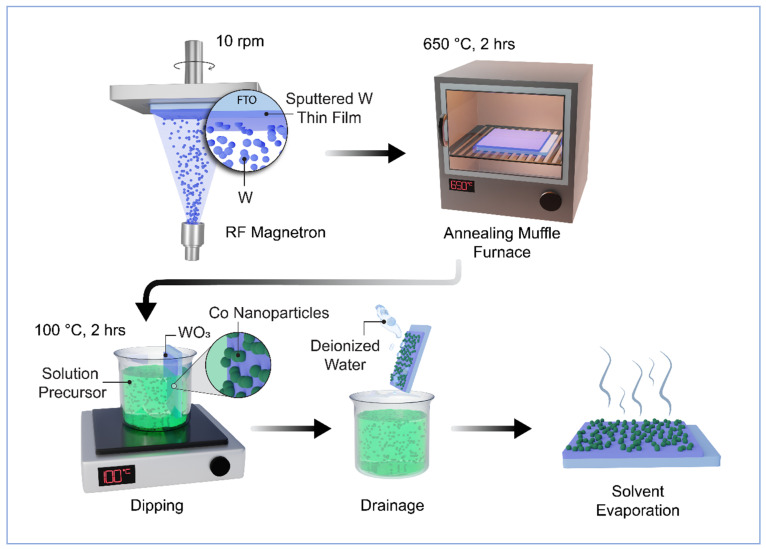
Schematic route diagram of Co/WO_3_ film fabrication.

**Figure 2 nanomaterials-14-00259-f002:**
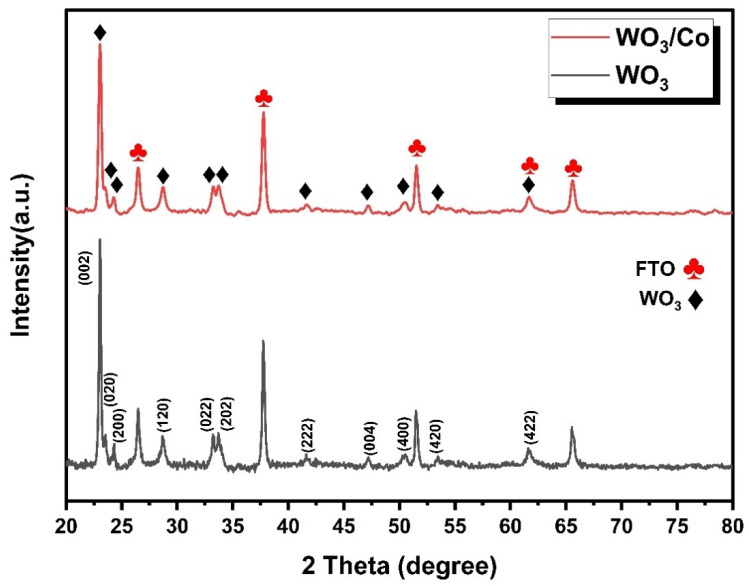
XRD patterns of pristine and Co/WO_3_ thin films.

**Figure 3 nanomaterials-14-00259-f003:**
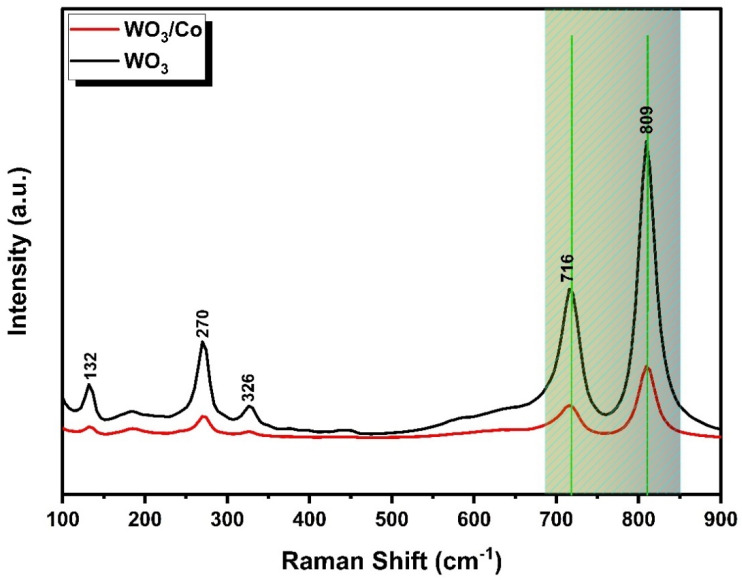
The Raman spectra of WO_3_ and Co/WO_3_ films.

**Figure 4 nanomaterials-14-00259-f004:**
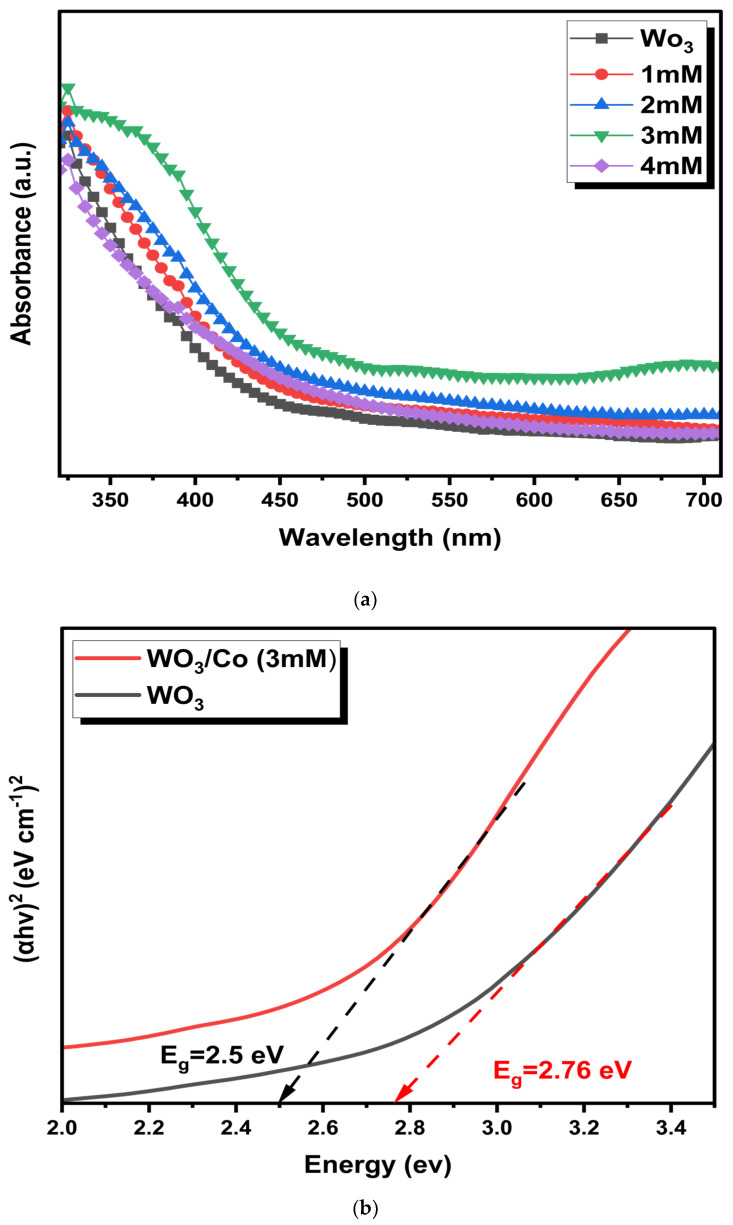
(**a**) Absorption spectra and (**b**) Tauc plot of WO_3_ and Co/WO_3_ thin films.

**Figure 5 nanomaterials-14-00259-f005:**
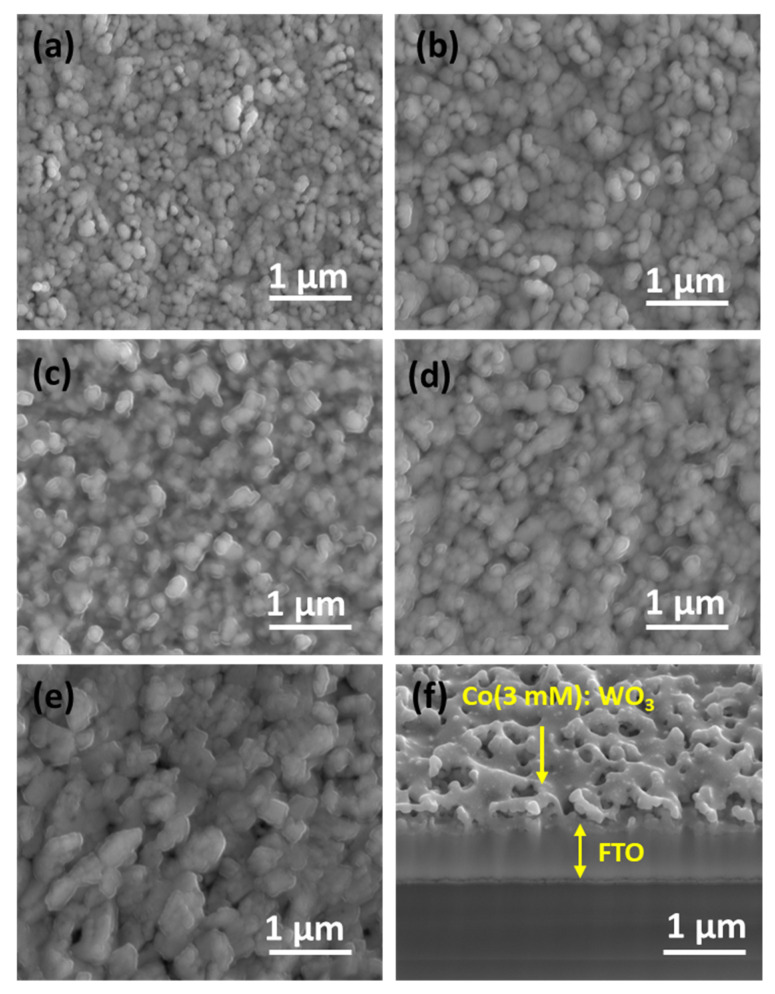
Surface FE-SEM images of WO_3_ films: (**a**) bare, (**b**) 1 mM, (**c**) 2 mM, (**d**) 3 mM, and (**e**) 4 mM concentrations. (**f**) A cross-section of the Co/WO_3_ (3 mM) film. (**g**) EDS spectrum of the Co/WO_3_ (3 mM) photoanode.

**Figure 6 nanomaterials-14-00259-f006:**
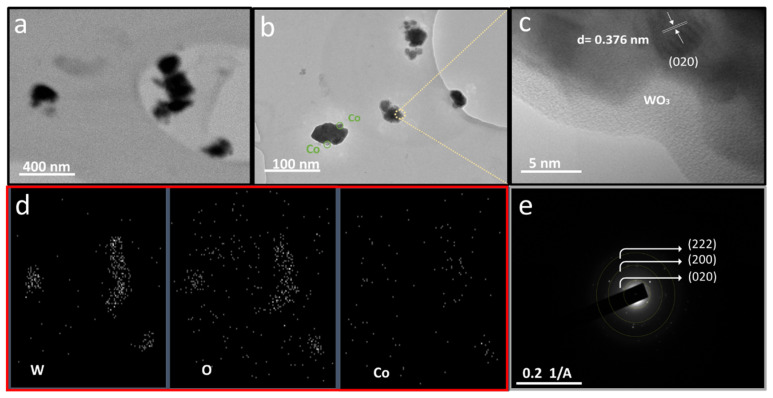
The morphologies of the Co/WO_3_ films (3 mM) were characterised using TEM: (**a**) low-magnification, (**b**) high-magnification, and (**c**) high-resolution TEM (HR-TEM); (**d**) STEM images of the Co/WO_3_ films (3 mM); and (**e**) SAED.

**Figure 7 nanomaterials-14-00259-f007:**
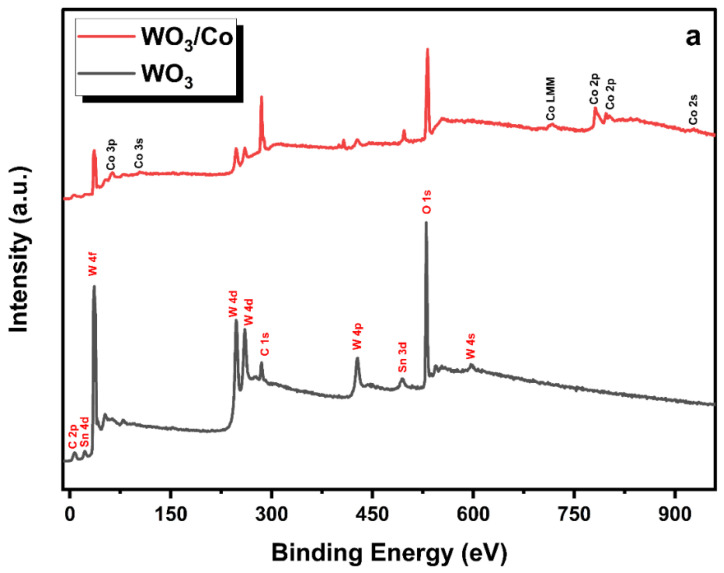
(**a**) XPS survey spectrum of the of WO_3_ and Co/WO_3_ (3 mM) thin films on FTO glass. Deconvoluted high-resolution XPS spectra of W 4f core level of (**b**) unloaded and (**c**) Co/WO_3_ (3 mM) film. (**d**,**e**) Deconvoluted high-resolution XPS spectra of O 1s for both samples. (**f**) Deconvoluted high-resolution XPS spectra of Co 2p core levels.

**Figure 8 nanomaterials-14-00259-f008:**
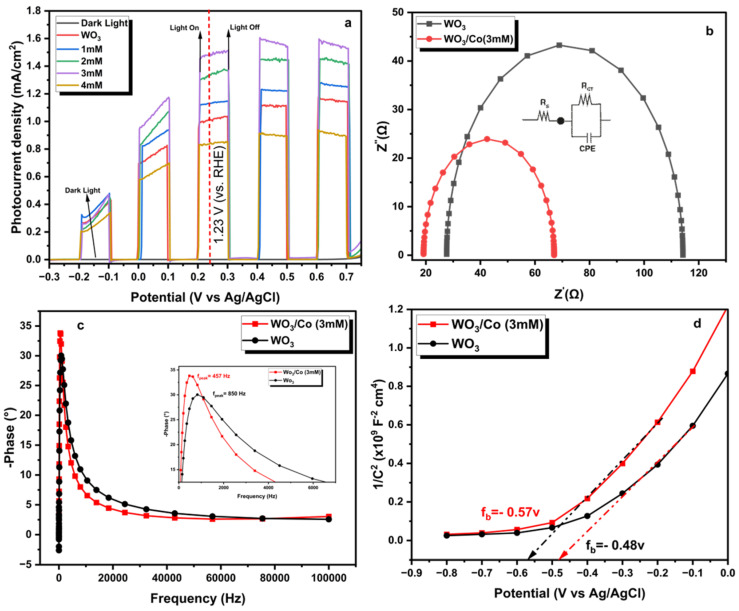
(**a**) Linear sweep voltammetry curves of the WO_3_ and Co/WO_3_ thin films under chopped light and dark illumination at a scan rate of 0.1 mV/s in a 1 M NaOH electrolyte (pH = 13.6). (**b**) Nyquist plot of the EIS measurements obtained for pure WO_3_ and Co/WO_3_ films at 1.23 V vs. RHE under light illumination, with the modelled circuit used to match the raw data in the inset. (**c**) Bode plots for both photoanodes. (**d**) Mott–Schottky data displaying the flat band potentials.

**Table 1 nanomaterials-14-00259-t001:** Summary of the photoelectrochemical performance of earlier reported photoanodes with different Co catalysts.

Photoanode	Synthesis Method	Electrolyte	Photocurrent Density	Reference Electrode	Reference
Ag–LaFeO_3_	spin coating	0.1 M NaOH	0.074 mA/cm^2^	0.6 V vs. RHE	[29]
Ni–LaFeO_3_	spin coating	0.1 M NaOH	0.066 mA/cm^2^	0.6 V vs. RHE	[30]
Co–BiVO_4_	electrochemical synthesis	0.1 M PBS	0.69 mA/cm^2^	1.23 V vs. RHE	[50]
Ag–ZnFe_2_O_4_	chemical water bath	0.5 M Na_2_SO_4_	0.91 mA/cm^2^	1.23 V vs. RHE	[51]
Mo–WO_3_	hydrothermal method	0.1 M Na_2_SO_4_	1.15 mA/cm^2^	0.8 V vs. Ag/AgCl	[10]
WO_3_	PVD/DC	0.1 M Na_2_SO_4_	0.9 mA/cm^2^	0.6 V vs. SCE	[49]
WO_3_	PVD/RF	1 M NaOH	1.020 mA/cm^2^	1.23 V vs. RHE	present work
Co-WO_3_	PVD/RF and thermal oxidation	1 M NaOH	1.485 mA/cm^2^	1.23 V vs. RHE	present work

## Data Availability

Data are available upon request to the corresponding author.

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
