# Peer review of "Enhanced Photoelectrochemical Performance Using Cobalt-Catalyst-Loaded PVD/RF-Engineered WO3 Photoelectrodes"

_nanomaterials, 2024, doi:10.3390/nano14030259_

Round 1

Reviewer 1 Report

Comments and Suggestions for Authors

Journal: Nanomaterials
Title: Enhanced Photoelectrochemical Performance by Co cocatalyst-loaded PVD/RF-Engineered WO3 Photoelectrodes.
Manuscript ID: nanomaterials-2799791
Review Comments: The authors fabricated WO3 thin films modified with Co nanoparticles on an FTO glass substrate and studied the performance of the electrode in photoelectrochemical water splitting. The work presented in the manuscript is within the scope of the journal. Hower the manuscript cannot be accepted in its current form. The authors may be advised to update their manuscript according to the comments listed below.

1.      What does "Co Cocatalyst" mean in the title? Is it a typo? The authors may consider updating the title of their manuscript.

2.      The authors mentioned that they deposited WO3 by PVD/RF sputtering; however, they deposited W by sputtering and then annealed the W film at a high temperature to oxidize the W to WO3. That is, they deposited WO3 on the FTO glass substrate by Sputtering following thermal oxidation instead of one-step reactive RF sputtering.

3.      The authors may label their pristine photoanode as WO3 instead of unloaded or untreated WO3 and Co nanoparticles loaded WO3 as Co-WO3 or Co-NPs/WO3 or Co-NPs-WO3 for referring to their electrodes consistently throughout the manuscript.

4.      The authors should update Table 1 with correct data according to the reference articles; especially, the reference electrodes and potentials are inconsistent in the table.

5.      In conclusion, the authors mentioned, “An eco-friendly, low-temperature PVD/RF method was employed to successfully fabricated pristine WO3 and Co-loaded WO3 thin films,” however, the oxidation of W to WO3 required high temperature.

6.      There are some typos in the manuscript, for example, “supported” at line 105.

7.      There are some incomplete sentences in the manuscript. For example, the sentence "In addition, 51 due to its non-toxicity, straightforward fabrication procedures, and exceptional stability 52 in acidic solution [12]." at lines 51-52.

Comments on the Quality of English Language

1.      There are some typos in the manuscript, for example, “supported” at line 105.

2.      There are some incomplete sentences in the manuscript. For example, the sentence "In addition, 51 due to its non-toxicity, straightforward fabrication procedures, and exceptional stability 52 in acidic solution [12]." at lines 51-52.

Author Response

Dear Reviewer 1,

I hope this message finds you well.

I am writing to inform you that my response to your comment regarding our manuscript has been compiled and is attached for your review. Please find the detailed response attached to this message. Your insightful feedback has been invaluable in enhancing the quality and rigor of our work, and we sincerely appreciate your guidance.

Thank you so much,

Best regards,

Mansour

Reviewer 2 Report

Comments and Suggestions for Authors

This manuscript reports the enhanced photoelectrochemical (PEC) performance by Co cocatalyst-loaded WO3 photoelectrodes. The results show that compared to pristine WO3 thin film photoanode, the photocurrent of as-prepared WO3/Co films shows a higher PEC activity with more than one fold of photocurrent density increase from 1.020 mA/cm2 to 1.485 mA/cm2 under simulated solar radiation. This manuscript is interesting by terms of the preparation methods, morphology and mechanism analysis. In addition, the conclusion can also be confirmed by the experimental results. The paper could be suitable and recommended for publication in the "Nanomaterials ", but some issues need clarification following a minor revision:

1.     Why did the author choose Co as co-catalyst instead of the more well-known Co-Pi and FeOOH? What is the advantage of this elemental co-catalyst? How is its cyclic stability?

2.     Table 1 should be expressed in a three-line format.

3.     No direct evidence suggests that surface Co modification can narrow the bandgap of WO3. It seems that the presence of Co would change the Fermi level of WO3. It had better provide First principles’ calculation results for clarifying the mechanism.

4.     The authors mentioned that the deposition of Co co-catalysts with a high concentration may lead to the aggregation of Co particles, affecting the PEC properties. This phenomenon is reasonable, but direct evidence needs to be provided (particle agglomeration wasn’t observed solely by low-magnification SEM images).

Author Response

Dear Reviewer 2,

I hope this message finds you well.

I am writing to inform you that my response to your comment regarding our manuscript has been compiled and is attached for your review. Please find the detailed response attached to this message. Your insightful feedback has been invaluable in enhancing the quality and rigor of our work, and we sincerely appreciate your guidance.

Thank you so much,

Best regards,

Mansour

Round 2

Reviewer 1 Report

Comments and Suggestions for Authors

The authors have updated the manuscript according to my comments; however, the Table 1 still has a problem since not all the data were measured against RHE as found in the reference articles. Author may update the table by removing "vs RHE" from the header adding a new column for reference electrodes.

Author Response

Dear Reviewer 1,

Thank you for your valuable feedback and for pointing out the inconsistency in Table 1 of our manuscript.  I appreciate your attention to detail and agree with your suggestion regarding the representation of the data in relation to the reference electrodes.

In response to your comment, I have revised Table 1 accordingly. I have removed "vs RHE" from the header to avoid any confusion. Additionally, I have introduced a new column specifically for the reference electrodes, as per your suggestion. This new column clearly indicates the type of reference electrode used in each study, ensuring that the data presented is accurate and aligned with the referenced articles.

I hope that these changes address your concerns and improve the clarity and accuracy of the information presented in our manuscript. Once again, I am grateful for your insightful comments and guidance.

Kind regards,

Mansour